# All-Directional DOA Estimation for Ultra-Wideband Regular Tetrahedral Array Using Wrapped PDoA

**DOI:** 10.3390/s22041532

**Published:** 2022-02-16

**Authors:** Jinglin Luo, Jingjing Zhang, Haidong Yang, Yisheng Guan

**Affiliations:** 1Foshan Nanhai Guangdong Technology University CNC Equipment Cooperative Innovation Institute, Foshan 528225, China; yanghd@gdut.edu.cn; 2School of Electromechanical Engineering, Guangdong University of Technology, Guangzhou 510006, China; 3School of Mechatronic Engineering and Automation, Foshan University, Foshan 528000, China; jingjing@fosu.edu.cn; 4Biomimetic and Intelligent Robotics Lab (BIRL), Guangdong University of Technology, Guangzhou 510006, China; ysguan@gdut.edu.cn

**Keywords:** ultra-wideband, regular tetrahedral array, DOA estimation, wrapped PDoA

## Abstract

In this paper, we proposed a Regular Tetrahedral Array (RTA) to cope with various types of sensors expected in Ultra-Wideband (UWB) localization requiring all-directional detection capability and high accuracy, such as indoor Internet-of-Things (IoT) devices at diverse locations, UAVs performing aerial navigation, collision avoidance and takeoff/landing guidance. The RTA is deployed with four synchronized Ultra-Wideband (UWB) transceivers on its vertexes and configured with arbitrary aperture. An all-directional DOA estimation algorithm using combined TDoA and wrapped PDoA was conducted. The 3D array RTA was decomposed into four planar subarrays solved as phased Uniform Circular Array (UCA) respectively. A new cost function based on geometric identical and variable neighborhood search strategy using TDoA information was proposed for ambiguity resolution. The results of simulation and numerical experiments demonstrated excellent performance of the proposed RTA and corresponding algorithm.

## 1. Introduction

All-directional detection for a single Ultra-wideband (UWB) source in an isotropic way become increasingly important. It is required in many UWB applications such as single anchor UWB localization system [1], UAVs collision avoidance [2,3,4], takeoff/ landing guidance [5,6], Internet-of-Things (IoT) devices, and vehicular-to-everything (V2X) communication [7]. Current antenna arrays applied in UWB localization, such as Uniform Circular Array (UCA), Uniform Linear Array (ULA) [8,9], have restrictions on their detection angle range in both azimuth and elevation.

Tiemann et al. [8] tested a UWB location system based on three synchronized UWB transceivers mounted on the helmet for supporting first responders through 3D location of fellows and victims in a low visibility environment. This antenna array consists of 2 ULAs perpendicular to each other, for measuring the Angle of Arrival (AoA) in the x-axis and y-axis respectively, using PDoA [10] of antennas. Similarly, Zhao et al. [9], tested a low-power, scalable and cm-accurate UWB location system, based on eight synchronized UWB transceivers mounted on a single PCB, four antennas in horizontal and other four antennas in vertical. The common imperfections of these two works are angle range limitation and fixed antenna spacing less than half-wavelength. The tight antenna spacing is designed for special frequencies that limit the flexibility of the antenna array. Furthermore, working at the centimeter band, mutual coupling between antennas disturbs received signals and degrades the DoA finding performance severely [11].

UCA is extensively utilized in the context of 2D direction finding due to its attractive advantages, including omnidirectional azimuth coverage, almost unchanged directional pattern, and about 90° elevation angle coverage [12,13,14]. The drawback of UCA is the sign ambiguity of elevation. To expand the range of signal detection overall azimuth and elevation angles, an array in spherical shape can be used for uniform and stable beamforming in all directions. By adding one more transceiver to the UCA, a Regular Tetrahedral Array (RTA) can be an available candidate. The Cramer-Rao Bound for direction finding of a tetrahedral array of isotropic sensors was studied in [15]. Using TDoA, Acres et al. [4] conducted a method of determining relative bearing and elevation for RTA. Based on euclidean distance and tetrahedron, Phalak et al. [16] presented a decentralized relative localization for Multi-Robot systems. However, neither TDoA nor distance measurement provides much lower localization accuracy than PDoA.

Expanding antenna spacing to larger than half-wavelength can not only reduce antenna coupling but also augment array aperture and improve AoA estimation accuracy [17]. However, the actual PDoA of signals cannot be obtained directly due to the phase wrapping problem [18,19], which can be solved by auxiliary measurements [20,21]. Ge et al. [1] develop a 3D single-anchor localization system based on UWB signals using an arbitrary geometry array. They also conducted an unwrapping PDoA method based on Fisher information matrix demanding a lot of computing power, even GPU in parallel computing. Xin et al. [18] reported an ambiguity resolution algorithm for passive 2-D source localization with a UCA. Their unwrapping PDoA is based on the estimation of the detected curve parameters using randomized Hough transform. The randomized Hough transform is usually used for curve detection in image processing, which also need extensive computing resource.

In this paper, we proposed a regular tetrahedral array (RTA), which deployed four synchronized Ultra-Wideband (UWB) transceivers on its vertexes and configured aperture larger than half-wavelength. Each UWB transceiver can identify the first path and provide an estimate of TDoA and PDoA at the same time. The RTA can be solved by decomposing this 3D array into four planar subarrays treated as phased UCA independently. Benefiting from the spatial complementarity of these four subarrays, a RTA not only get the capability of detecting signal source in all direction but also get redundancy when antenna failure or shield [22]. To cope with wrapped PDoA caused by larger antenna spacing, we proposed an ambiguity resolution algorithm based on geometric identical, which consists of two parts: one is a new cost function based on identical source direction vectors (SDV) that estimated by four subarrays and another is ambiguity integer search strategy. To improve the robustness of the algorithm, we designed a voting mechanism for filtering noised information to get accurate SDV results. The proposed ambiguity resolution algorithm improve estimation accuracy and reduce computing resource consumption. Meanwhile, the ambiguity resolution algorithm allows more flexibility for the selection of an array radius and has further applications for unambiguous direction finding in a very wide frequency band.

The remainder of this paper is organized as follows: Section 2 is a coarse SDV estimation only using TDoA information. This coarse estimate is utilized to solve the sign ambiguity of elevation when using phased UCA. Section 3 conducted SDV using wrapped PDoA and TDoA. In this section, we decomposed the RTA into four UCA subarrays. Utilizing the identical of SDV estimated by each subarray, the ambiguity resolution algorithm was conducted. The performance of the proposed algorithm was evaluated in Section 4. A conclusion was made at the end of this paper in Section 5.

## 2. Coarse Estimate for RTA Using TDOA

Taken antenna A as a reference, note TDoA measurements $\tau ={\left(\tau_{BA},\tau_{CA},\tau_{DA}\right)}^{T}$, and PDoA measurements $\phi ={\left(\phi_{BA},\phi_{CA},\phi_{DA}\right)}^{T}$.

Figure 1 illustrates the geometrical shape of a RTA. *r* is the radial distance from the center of the triangular base to the antenna B, C, and D. *h* is the vertical height of the antenna A above the center of the triangular base. The coordinate system $O_{xyz}$ is located at the centroid of the triangle $\Delta_{BCD}$, which is right-handed with z positive upwards and x positive toward antenna B. In a RTA $h=\sqrt{2}r$.

Assuming there is a single signal source in the far field, S is the wavefront plane of the signal. When wavefront plane S passes through antenna A, S can be described as $v_{1}x+v_{2}y+v_{3}(z-h)=0$, the unit normal vector of the plane S is $n=v_{TDoA}={\left[\frac{v_{1}}{\sqrt{v_{1}{}^{2}+v_{2}{}^{2}+v_{3}{}^{2}}},\frac{v_{2}}{\sqrt{v_{1}{}^{2}+v_{2}{}^{2}+v_{3}{}^{2}}},\frac{v_{3}}{\sqrt{v_{1}{}^{2}+v_{2}{}^{2}+v_{3}{}^{2}}}\right]}^{T}$, meanwhile, $v_{TDoA}$ is a source direction vector (SDV). In this paper, we present the direction of arrival using unit vector SDV for easier calculation and no gimbal lock. 

The distance between antenna B, C, D, and plane S are:(1)$$d_{BA}=\tau_{BA}c=\frac{\frac{2\sqrt{2}}{3}rv_{1}-\frac{4}{3}rv_{3}}{\sqrt{v_{1}{}^{2}+v_{2}{}^{2}+v_{3}{}^{2}}},$$
(2)$$d_{CA}=\tau_{CA}c=\frac{-\frac{\sqrt{2}}{3}rv_{1}-\frac{\sqrt{6}}{3}rv_{2}-\frac{4}{3}rv_{3}}{\sqrt{v_{1}{}^{2}+v_{2}{}^{2}+v_{3}{}^{2}}},$$
(3)$$d_{DA}=\tau_{DA}c=\frac{-\frac{\sqrt{2}}{3}rv_{1}+\frac{\sqrt{6}}{3}rv_{2}-\frac{4}{3}rv_{3}}{\sqrt{v_{1}{}^{2}+v_{2}{}^{2}+v_{3}{}^{2}}},$$
where, c is the propagation speed of the electromagnetic wave in the air. Rearrange Equations (1)–(3), we get equation:(4)$$Av_{TDoA}=b,$$
where,
$$A=\left[\begin{array}{ccc} 2\sqrt{2}r/3 & 0 & -4r/3\\ -\sqrt{2}r/3 & -\sqrt{6}r/3 & -4r/3\\ -\sqrt{2}r/3 & \sqrt{6}r/3 & -4r/3 \end{array}\right],\;b=\left[\begin{array}{c} \tau_{BA}c\\ \tau_{CA}c\\ \tau_{DA}c \end{array}\right].$$

A is invertible matrix, the solution of SDV can be obtained as follows:(5)$$v_{TDoA}=A^{-1}b$$

Since TDoA measurements are always polluted by clock drift, unbalanced antenna delay, and ADC sampling error, therefore, $v_{TDoA}$ suffer from noise and errors. We use it as a coarse estimate to solve the sign ambiguity of elevation when using phased UCA in Section 3 and standby results in the event of PDoA estimation failure.

## 3. Combined TDoA and Wrapped PDoA for RTA 

UWB transceiver can measure carrier phase more precisely than time-of-flight, the typical error value is less than 3°, which corresponds to 0.06 cm at $f_{c}=3.9936\;\mathrm{GHz}$. PDoA measurements error is about 1600 times smaller than TDoA measurements error [1]. For higher accuracy, we need to solve RTA using PDoA.

### 3.1. Wrapped PDoA for RTA

#### 3.1.1. Spatial Subarray Decompose of RTA

A RTA consists of four regular triangular subarrays, which can be treated as phased UCAs for solving 2D-AoA problems independently. Figure 2 depicts the decomposition of a tetrahedral and the spatial relationship between UCA subarrays and RTA.

In Figure 2, $O_{\Delta BCD},O_{\Delta ABD},O_{\Delta ADC},O_{\Delta ACB}$ are centroid of triangle $\Delta_{BCD},\Delta_{ABD},\Delta_{ADC},\Delta_{ACB}$ respectively, or Δ1,Δ2,Δ3,Δ4 for short. CO=O;x,y,z is the coordinate system in global and located at the original point O. COΔ=[OΔ;e1,e2,e3] is the right-handed coordinate system located at the centroid of a triangle, $e_{3}$ is perpendicular to the triangle surface and $e_{1}$ is towards the antenna numbered as 1 in the subarray. The transform matrix from COΔ to CO is ROΔO=e1,e2,e3. In each triangle, we can estimate an SDV. Theoretically, four estimated SDVs are all identical $v_{PDoA,\Delta 1}=v_{PDoA,\Delta 2}=v_{PDoA,\Delta 3}=v_{PDoA,\Delta 4}$, although these four subarrays are in different spatial positions. Moreover, these four subarrays share the same PDoA measurements and TDoA measurements, which give us a chance to unwrap PDoA ambiguity numerically.

#### 3.1.2. Solve Phased UCA Utilizing Fourier Analysis

DoA estimation of a phased UCA is well developed in both theory and technique. To avoid eigenvalue calculation, the algorithm here we used to solve the DOA estimation problem in the UCA subarray is based on the Fourier analysis of the phase around the circular aperture [13,14,23].

Consider a UCA with N identical elements illuminated by a single far-field source. Consider a circular aperture located at $\left(r,\pi /2,\varphi \right)$, in the spherical coordinate system of $\left(r,\theta ,\varphi \right)$, as shown in Figure 3. $\tilde{\phi }\left(\varphi \right)$ is the continuous curve of actual phase difference $\phi \left(i,1\right)$. The period of $\tilde{\phi }\left(\varphi \right)$ is 2π. The purple elliptic $E_{S,\phi }$ is the projection of the aperture circle $O_{\Delta }$ on the plane S. $\tilde{\phi }\left(\varphi \right)\lambda /2\pi$ is the distance between a point of aperture circle $O_{\Delta }$ and its projection point in elliptic $E_{S,\phi }$. The intersection line of elliptic $E_{S,\phi }$ and plane *S* is in blue. The normal vector of $E_{S,\phi }$ is $n_{S}=v_{PDoA,\Delta }$.

The phase of the electromagnetic field of an incident wave from $\left(\theta ,\varphi \right)$ can be written as
(6)$$\Phi \left(\varphi_{i}\right)=\frac{2\pi }{\lambda }r\sin\theta \cos\left(\varphi -\varphi_{i}\right)+\Phi_{0}$$
where the azimuth angle φ∈[0,2π) is measured counter-clockwise from the $e_{1}$-axis and the elevation angle θ∈[0,π) is measured down from the $e_{3}$-axis and is the wavelength λ. Antennas were located counter-clockwise around the circular, and numbered 1 to M. Antenna azimuth position are $\varphi_{i}=\frac{\pi \left(i-1\right)}{M},i=1,2,\ldots ,M$, where i is antenna number in the subarray, M is the total antenna number in the subarray, particularly, in a regular triangle subarray M=3. $\Phi_{0}$ is a constant and represents the initial phase of the incident wave, which can be removed by the phase difference.

Take antenna 1 as a reference, the actual phase difference between antenna i and 1 can be described as the following equation:(7)$$\begin{array}{ll} \phi \left(i,1\right) & =\Phi \left(\varphi_{i}\right)-\Phi \left(\varphi_{1}\right)\\ & =\frac{4\pi r}{\lambda }\sin\theta \sin\left(\frac{\pi \left(i-1\right)}{M}\right)\sin\left(\varphi -\frac{\pi \left(i-1\right)}{M}\right) \end{array}$$

When r>λ/2 the phase range may exceed 2π, which leads to an ambiguity in determining the direction of the incident wave plane S. Therefore, the actual phase difference ϕi,1 consists of two parts, namely, measured phase difference ϕ0i,1 and ambiguity part 2πNi,1,
(8)ϕi,1=ϕ0i,1+2πNi,1,
where ϕ0i,1∈(−π,π]. Ni,1∈ℤ is ambiguity integers that we need solve. The first order Fourier series coefficient of ϕi,1 is
(9)Ψ1=2πM∑i=1Mϕi,1expj2πi−1M,

According to dependence relationship, the elevation θ and azimuth φ are as follows:(10)$$\theta =\sin^{-1}\left(\frac{\lambda }{2\pi^{2}r}\left|\Psi 1\right|\right),$$
(11)$$\varphi =\arg\left(\Psi 1\right),$$
where, $\left|\Psi 1\right|$ denotes modulus of a complex number Ψ1. $\arg\left(\Psi 1\right)$ is the angle of the complex number Ψ1.

Then we get the SDV in coordinate COΔ:(12)vOΔPDoA=cosφsinθ,sinφsinθ,signΔcosθT,
where $sign_{\Delta }=sign\left(v_{TDoA}{}^{T}e_{3}\right)$ is the sign of elevation estimated in a triangle subarray, which is dependent on the angle between coarse estimation $v_{TDoA}$ and $e_{3}$.

Using coordinate transform equation:(13)vOPDoA=ROΔOvOΔPDoA,

Now we get a direction vector in a coordinate CO estimated by a triangle subarray, without loss of generality, one can calculate SDV in any subarray easily.

### 3.2. Ambiguity Resolution Algorithm

It is well known that high AOA estimation accuracy can be obtained for large apertures. However, when r>λ/2, the phase range may exceed 2π, which leads to an ambiguity in determining the direction of the incident wave. We continue to adopt the particular geometric properties of the RTA for ambiguity resolution.

#### 3.2.1. Geometric Identical Cost Function

We proposed a brand new cost function for ambiguity resolution based on the geometric identical of subarrays’ SDVs. As Figure 2 shows, SDVs estimated by four different subarrays are identical. Without loss of generality, any two adjacent subarrays, say, Δ2 and Δ3, they have common antennas A and D, and common reference antenna A. vOPDoA,Δ2 and vOPDoA,Δ3 are SDVs estimated in subarrays Δ2 and Δ3, respectively, using Equations (7)–(13). The cost function is written as:(14)M23=1−vOPDoA,Δ2TvOPDoA,Δ3,
where the footnote of $M_{23}$ 23 means subarray Δ2 versus subarray Δ3. $M_{23}$ is a scalar value, $M_{23}\in \left[0,2\right]$ describing error between vOPDoA,Δ2 and vOPDoA,Δ3. Distinguishing from current cost function based on Fourier inverse transform [14] or mapping tetrahedral volume [18], which estimating ambiguity integers firstly and then calculating DoA subsequently, unavoidable large rounding errors, our cost function is based on examining SDVs directly. From Equation (9), we know $M_{23}$ is a scalar filed with 6 independent variables. Given PDoA measurements $\phi_{0}\left(B,A\right),\phi_{0}\left(C,A\right),\phi_{0}\left(D,A\right)$ as a priori knowledge. $M_{23}$ can be reduced as discrete three dimensions, that is $M_{23}\left(N_{BA},N_{CA},N_{DA}\right)$.

The ambiguity resolution problem transforms into an optimization problem on discrete feasible set $N=\left\{N_{BA},N_{CA},N_{DA}\right\}$. The constraint of the aforementioned optimization problem can be extracted from Equations (9) and (10). The first order Fourier series coefficient Ψ1 is a complex function of $N=\left\{N_{BA},N_{CA},N_{DA}\right\}$, depending on geometric constraints $0\leq \sin\theta <1,\theta \in \left[0,\pi \right)$ we get,
(15)$$\left|\Psi 1\left(N_{BA},N_{CA}\right)\right|<\frac{2\pi^{2}r}{\lambda },$$
(16)$$\left|\Psi 1\left(N_{DA},N_{CA}\right)\right|<\frac{2\pi^{2}r}{\lambda },$$

Given subarray radius r=0.12 m and carrier frequency $f_{c}=3.9936\;\mathrm{GHz}$, the corresponding feasible set is illustrated in Figure 4. The red point is target ambiguity integers and other points are feasible set with color indicating cost value. The points that disobeyed Equations (15) and (16) are hidden.

In order to further study the geometric characteristic of cost function intuitively, the cost value is depicted in a meshed surface projected on $N_{CA},N_{DA}$ plane in Figure 5. According to geometric symmetry in the RTA, the geometric characteristic of cost value projected on $N_{BA},N_{DA}$ plane is similar to it on $N_{CA},N_{DA}$ plane. Therefore, only one map is depicted here.

As shown in Figure 5a, the cost function is non-convex because it is discrete and multiple local extreme points [24]. Therefore, ambiguity resolution problem cannot be solved by the gradient descent method. To make the cost value of the target point and other local extreme points more obvious, Figure 5a was redrew by $\log_{10}\left(M\right)$ in Figure 5b. The cost value at target points is in approximate $10^{-7}$ orders. While, the cost value of other local extreme points are in $10^{-2}$ orders. An error tolerance threshold ε>0 can distinguish the target point from other local extreme points.

#### 3.2.2. Ambiguity Integer Search Strategy

Because the cost function is non-convex and cannot search ambiguity integers by the gradient descent method, a good initial value and search strategy are key factors for search success and rapid goal. Assuming the noise of TDoA and PDoA measurements are AWGN with zero means. $n_{\tau }∼N\left(0,\sigma_{\tau }\right)$ and $n_{\phi }∼N\left(0,\sigma_{\phi }\right)$, and the noises of each receiver are independent. TDoA information is ideal auxiliary measurements to solve the phase wrapping problem because rounding operation is an estimate of ambiguity integers. According to the probability distribution $n_{\tau }∼N\left(0,\sigma_{\tau }\right)$, the probability of catching the goal is higher when the search area is closer to the initial value, Therefore, for the discrete search area, a small and tight neighborhood of initial value is more favorable than a full-range search area for a rapid goal.

##### Initial Value

Essentially, the bond between time difference and phase difference is the distance between antenna *i* and the wavefront plane S, that is
(17)$$\tau_{i,1}c=-\frac{\lambda }{2\pi }\phi \left(i,1\right),$$
where $\tau_{i,1}$ is the time difference between antenna *i* and 1. $\phi \left(i,1\right)$ is the actual phase difference between antenna *i* and 1. Substitute Equation (8) in Equation (17), and rearrange, we get
(18)$$N_{i,1}=-\frac{\tau_{i,1}c}{\lambda }-\frac{\phi_{0}\left(i,1\right)}{2\pi },$$

Due to $\phi_{0}\left(i,1\right)\in (-\pi ,\pi ]$, an estimate of ambiguity integer initial values are as follows:(19)$${\hat{N}}_{i,1}=ceil(-\frac{\tau_{i,1}c}{\lambda }-\frac{1}{2}),$$
where, ceil(x) denotes the least integer greater than or equal to x. A set of reasonable initial values of ambiguity integers can be guessed from the TDoA measurements. ${\hat{N}}_{BA}=ceil\left(-d_{BA}/\lambda -1/2\right)$, ${\hat{N}}_{CA}=ceil\left(-d_{CA}/\lambda -1/2\right)$, ${\hat{N}}_{DA}=ceil\left(-d_{DA}/\lambda -1/2\right)$, ${\hat{N}}_{DB}=ceil\left(-\left(d_{DA}-d_{BA}\right)/\lambda -1/2\right)$, ${\hat{N}}_{CB}=ceil\left(-\left(d_{CA}-d_{BA}\right)/\lambda -1/2\right)$.

##### Variable Neighborhood Search

Figure 6 illustrates the variable neighborhood search strategy in an arbitrary subarray. According to geometric symmetry in the RTA, only one map is illustrated here. The curve $\tilde{\phi }\left(\varphi \right)$ in red is the actual phase difference curve. Black points at $\frac{2\pi }{3}$ and $\frac{4\pi }{3}$ are phase differences consist of ambiguity integer initial values estimated by TDoA measurements. The curve ${\tilde{\phi }}_{TDoA}\left(\varphi \right)$ in the black is the phase difference curve estimated by TDoA. Green points are 1-neighborhood search points, which are 1 step or 2π away from initial values. The curve ${\tilde{\phi }}_{Search}\left(\varphi \right)$ in green is 1-neighborhood upper and lower search boundary. The green arrows indicate the expanding direction of search points, the expanding step is 2π.

We proposed a variable neighborhood search strategy, which starts the search from ambiguity integer initial values estimated from TDoA measurements. We denoted the initial value sets as ${\hat{N}}_{s}=\left\{{\hat{N}}_{BA},{\hat{N}}_{CA},{\hat{N}}_{DA}\right\}$. ${\hat{N}}_{DB}$, ${\hat{N}}_{CB}$ can be described by linear combination of ${\hat{N}}_{s}$, that reduce time complexity from $O\left(N^{5}\right)$ to $O\left(N^{3}\right)$.

The variable neighborhood search strategy as follows:

Firstly, try the initial value ${\hat{N}}_{s}$, If not catch the goal, move on 1- neighborhood traversal search, which expanding search area to:(20)$$\begin{array}{ll} N_{s} & ={\hat{N}}_{s}\pm 1\\ & =\left\{\left[{\hat{N}}_{BA}-1,{\hat{N}}_{BA}+1\right],\left[{\hat{N}}_{CA}-1,{\hat{N}}_{CA}+1\right],\left[{\hat{N}}_{DA}-1,{\hat{N}}_{DA}+1\right]\right\} \end{array}$$

If does not catch the goal either, move on 2-neighborhood traversal search, which expanding search area to:(21)$$\begin{array}{ll} N_{s} & ={\hat{N}}_{s}\pm 2\\ & =\left\{\left[{\hat{N}}_{BA}-2,{\hat{N}}_{BA}+2\right],\left[{\hat{N}}_{CA}-2,{\hat{N}}_{CA}+2\right],\left[{\hat{N}}_{DA}-2,{\hat{N}}_{DA}+2\right]\right\} \end{array}$$

If does not catch the goal either, move on and on, until catch the goal or reach the $N_{\max}$. In the worst case that search in $N_{\max}$, the computational complexity of our proposed algorithm is $O\left(4\left(M-1\right){\left(2N_{\max}+1\right)}^{M}\right)$, M=3

#### 3.2.3. Spatial Subarray Vote Mechanism

From four subarrays in a RTA, four SDV estimations are accumulated in a matrix $V_{PDoA}=\left[\begin{array}{cccc} v_{PDoA,\Delta 1} & v_{PDoA,\Delta 2} & v_{PDoA,\Delta 3} & v_{PDoA,\Delta 4} \end{array}\right]$. A cost function in matrix form can be written as
(22)$$M=I_{4\times 4}-{\left(V_{PDoA}\right)}^{T}V_{PDoA},$$

Specifically,
(23)$$M=\left[\begin{array}{cccc} 0 & M_{12} & M_{13} & M_{14}\\ M_{12} & 0 & M_{23} & M_{24}\\ M_{13} & M_{23} & 0 & M_{34}\\ M_{14} & M_{24} & M_{34} & 0 \end{array}\right]$$

The component $M_{ij}$ is the cost function of each adjacent subarrays. We designed a mechanism for deciding whether the result SDVs are acceptable. This mechanism is called spatial subarray voting. Take an array as a ballot box $V_{vote}=\left[V_{12},V_{13},V_{14},V_{23},V_{24},V_{34}\right]$. Vote counting using an error tolerance ε>0.
(24)$$\begin{array}{l} V_{ij}=\left\{\begin{array}{c} 1,\;M_{ij}\leq \varepsilon \\ 0,\;\mathrm{others} \end{array}\right.\;,\\ \left(i\neq j\;i=1,2,3\;j=2,3,4\right) \end{array}$$

Typically, ε is machine precision of a computer or a relaxation precision considering.

When all possible $N_{s}$ are searched over, we will know the total votes and get out a DoA estimation. On the other hand, if there are no votes at all, a bigger radius neighborhood search area is expanded for next around search, until getting enough votes or approaching the maximum search boundary.

The final decision depends on the total count of votes, $D=\sum V_{ij}$. Theoretically, all four SDVs should be identical, in other words D=6. Affecting by lower accuracy of TDoA measurements, when elevation sign inverse or target missing in some subarrays occurrence, we need relax final decision condition to D≥3. If there are enough votes, the result is estimated by corresponding subarrays.
(25)$$\begin{array}{l} v_{PDoA}=\frac{1}{2D}\sum_{i,j}^{}V_{ij}\left(v_{PDoA,\Delta i}+v_{PDoA,\Delta j}\right),\\ \left(i\neq j\;i=1,2,3\;j=2,3,4\right) \end{array}$$

While in the situation of no vote at all, the final result should be the SDV estimated by TDoA, namely $v_{TDoA}$.

## 4. Simulation Results

To demonstrate the effectiveness and performance of the proposed algorithm, simulation and numerical experiments were conducted.

Assuming the noise of TDoA and PDoA measurements are AWGN with zero means, $n_{\tau }∼N\left(0,\sigma_{\tau }\right)$ and $n_{\phi }∼N\left(0,\sigma_{\phi }\right)$, and the noises of each receiver are independent. $\sigma_{\tau }$ is TDoA measurement error, $\sigma_{\phi }$ is PDoA measurement error. For examining the accuracy of proposed method, a series experiments were conducted. A set of rand SDVs covering all spherical surface were used as reference. And a set of RTA with different ratio r/λ estimated DoA of the reference SDV in different SNR conditions.

When $\mathrm{SDV}={\left(0.7001,0.7001,0.1400\right)}^{T}$, Figure 7 shows the accuracy comparison of coarse estimation using TDoA and proposed method.

In different SNR conditions, the error tolerance ε was set to $10^{-6}$ when SNR = 40 dB and set to $10^{-4}$ when SNR = 20 dB. The RMS error of angles using both TDoA and the proposed wrapped PDoA reduce when SNR increases.

Given subarray radius r=0.12 m and carrier frequency $f_{c}=3.9936\;\mathrm{GHz}$, the wavelength is λ=0.075120m. When SNR = 20 dB, RMS error of φ is about 3.165° using TDoA and 0.0942° using proposed method, while, RMS error of θ is about 1.607° using TDoA and 0.1981° using proposed method. When SNR = 40 dB, RMS error of φ is about 0.3087° using TDoA and 0.017° using proposed method, while, RMS error of θ is about 0.1888° using TDoA and 0.0379° using proposed method. The accuracy of our proposed algorithm is approaching closely to CRLB with different r/λ ratio.

When configured with different ratio r/λ, the corresponding angle RMS errors drop-down according larger ratio r/λ.

For testing the performance of proposed search strategy, we measured search steps of three different search strategies in the same condition. Given $\mathrm{SDV}={\left(0.7001,0.7001,0.1400\right)}^{T}$, subarray radius r=0.12m and carrier frequency $f_{c}=3.9936\;\mathrm{GHz}$. The size of the corresponding feasible set is ${\left(2N_{\max}+1\right)}^{3}=729$, where $N_{\max}=ceil\left(\sqrt{3}r/\lambda +1/2\right)=4$. The results were drew in Figure 8. Comparing different search strategies, the proposed method, which adopted TDoA initial values and variable neighborhood search, demonstrated excellent performance.

Illustrated in Figure 8, the x-axis is $\sigma_{\tau }c/\lambda$, which describes coarse estimation deviation from the reference. The y-axis is search steps starting from the initial value to catching the goal. The curve with red color is search steps adopting TDoA initial value and fixed search area ${\left[-4,4\right]}^{3}$. The curve in purple is search steps adopting zeros initial value and fixed search area ${\left[-4,4\right]}^{3}$. The curve in blue is search steps adopting TDoA initial value and variable neighborhood search, which proposed in this paper. Zeros initial value and the fixed search area is a conventional strategy that is used in current ambiguity resolution widely, which no need of any prior knowledge. No matter what the ratio value is, the search steps is about 190 and almost keep the same. We use it as a baseline strategy for evaluating others. When starting with TDoA initial value, only one-step is needed when the ratio $\sigma_{\tau }c/\lambda$ is low, but search steps rising quickly and maintaining at 140–180 closing to baseline strategy. It is obvious that the proposed method has advantages, that only one step to catch the goal when $\sigma_{\tau }c/\lambda \leq 0.15$ and about 20 steps when $0.2\leq \sigma_{\tau }c/\lambda <1$. When $\sigma_{\tau }c/\lambda \geq 1$, search steps raise high about 120 and approach to baseline strategy.

To find how the time difference SNR and the phase difference SNR affecting search success or not together of different ratio r/λ, another series experiments were conducted, in the same condition of RMS error examining. The error tolerance ε was set as $10^{-4}$ to adapt to low SNR conditions. Figure 9 shows numerical experiments results of the boundary of searching success for the different ratio r/λ.

The region of the upper and right sides of that boundary is the searching success region, which means if phase difference SNR and time difference SNR are both higher than require conditions the proposed method would catch the goal successfully after certain search steps. While, on the other hand, the lower and left corner of this map means unsuccessful search. From Figure 9, we know that the larger ratio r/λ, the more depending on TDoA informations and require higher time difference SNR to catch the goal successfully.

## 5. Conclusions

In this paper, we proposed a regular tetrahedral array (RTA), which deployed four synchronized Ultra-wideband (UWB) transceivers on its vertexes and configured with arbitrary aperture. An all-directional DOA estimation algorithm using combined TDoA and wrapped PDoA was conducted. A new cost function based on geometric identical and variable neighborhood search strategy using TDoA information was proposed for ambiguity resolution. Simulation and numerical experimentation results demonstrated excellent performance of the proposed RTA and corresponding algorithm.

When SNR = 20 dB, Using proposed method, the azimuth angle RMS error is about 0.0942° and elevation angle RMS error is about 0.1981°. The accuracy of proposed method is at least 18 times higher than the method using only TDoA. Comparing different search strategies, the proposed method adopting TDoA initial value and variable neighborhood search strategy demonstrated excellent performance. When $0.2\leq \sigma_{\tau }c/\lambda <1$, the search steps are about 20. When $\sigma_{\tau }c/\lambda \leq 0.15$, the search goal catches at the very first step. At last, the boundarys of searching success for the different ratio r/λ were found from the results of numerical experiments.

## Figures and Tables

**Figure 1 sensors-22-01532-f001:**
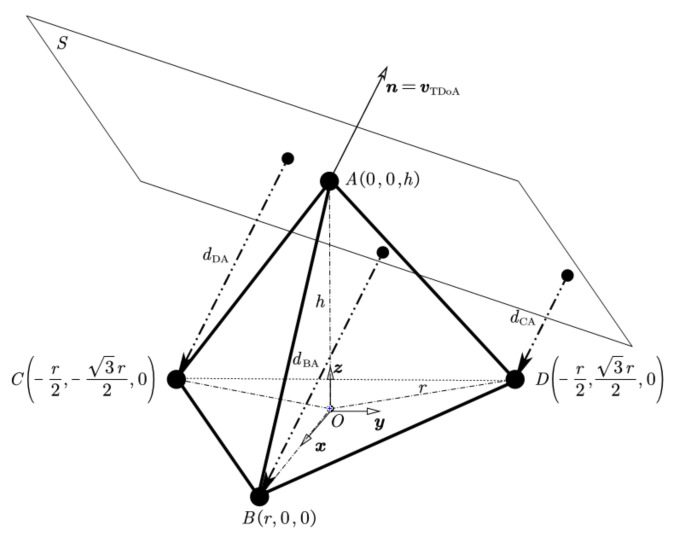
The geometrical shape of a regular tetrahedral array (RTA).

**Figure 2 sensors-22-01532-f002:**
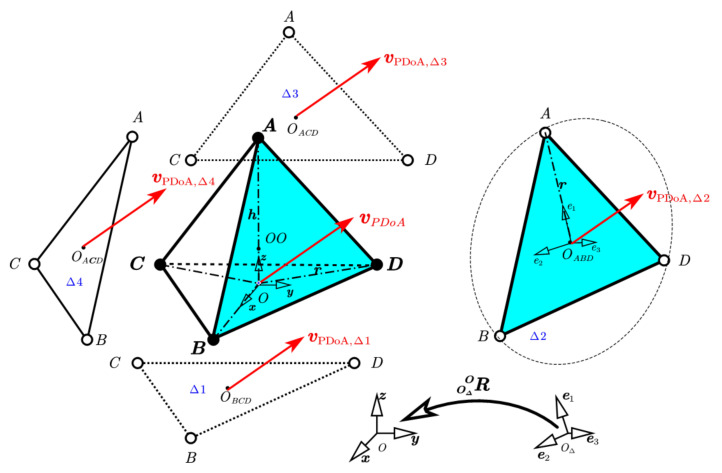
Tetrahedral decomposition and spatial relationship.

**Figure 3 sensors-22-01532-f003:**
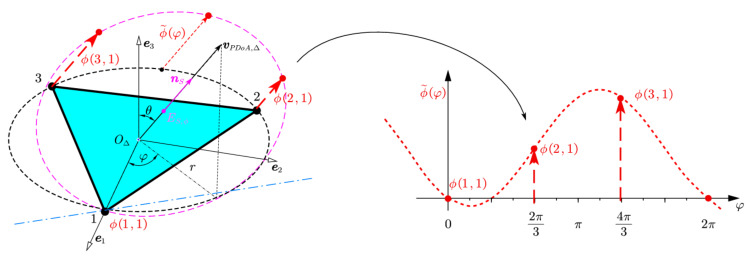
The visualization of actual phase difference $\phi \left(i,1\right)$.

**Figure 4 sensors-22-01532-f004:**
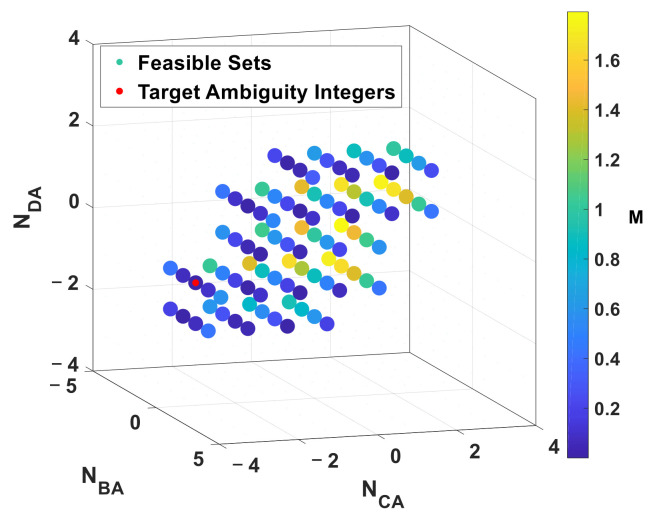
The Feasible Set of the proposed cost function in 3D. The red point is target ambiguity integers and other points (circle with color in image) are feasible set with color indicating cost value.

**Figure 5 sensors-22-01532-f005:**
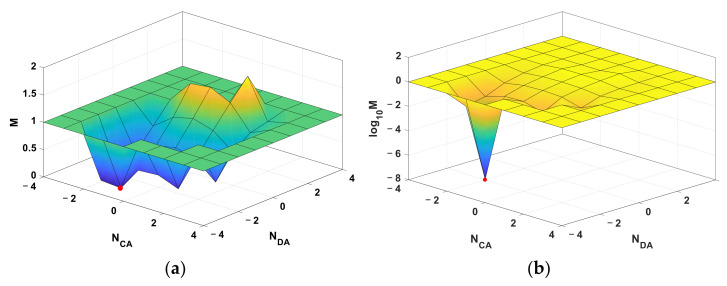
Visualization of cost value projected on $N_{CA},N_{DA}$ plane. (**a**) Meshed surface of cost value *M*, (**b**) Meshed surface of $\log_{10}\left(M\right)$. The red point is cost value of target ambiguity integers.

**Figure 6 sensors-22-01532-f006:**
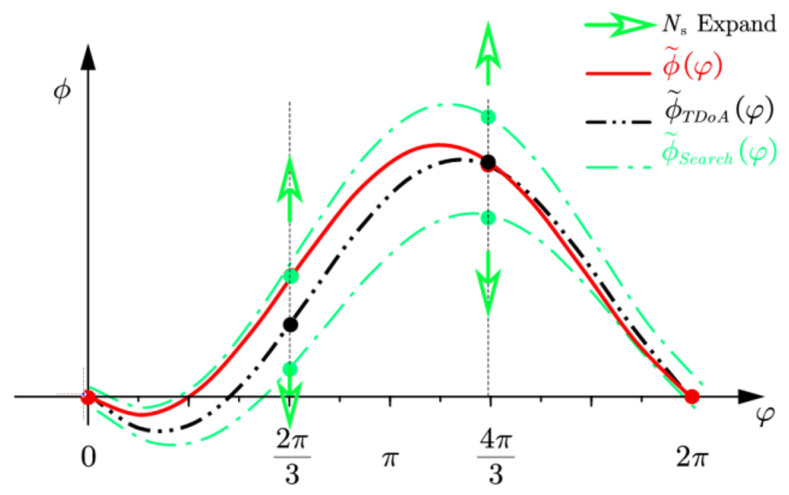
Variable neighborhood search.

**Figure 7 sensors-22-01532-f007:**
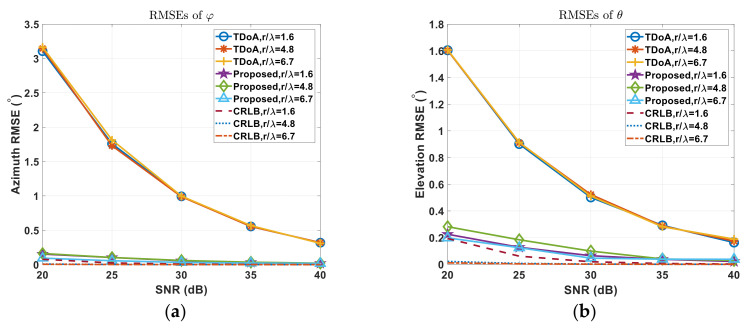
(**a**) RMS Errors of Azimuth angle φ and (**b**) RMS Errors of Elevation angle θ by TDoA only, proposed method and cramer-rao lower bound(CRLB) in different ratio r/λ.

**Figure 8 sensors-22-01532-f008:**
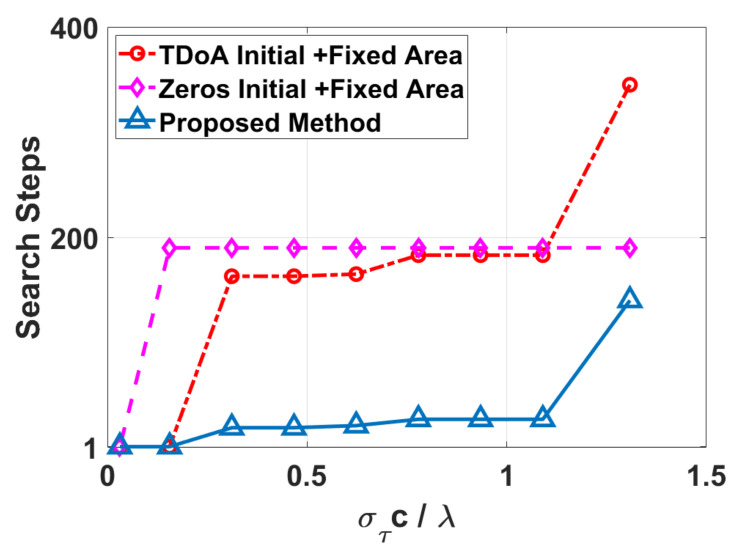
Search Steps of different search strategies.

**Figure 9 sensors-22-01532-f009:**
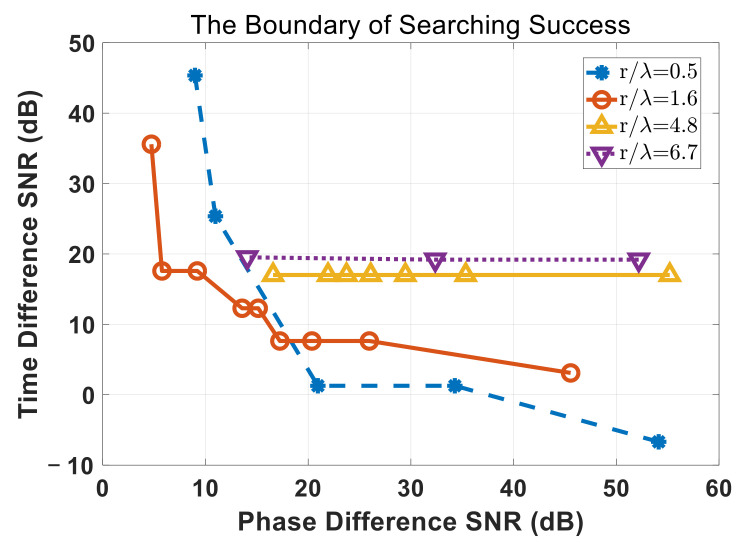
The boundary of searching success of different ratio r/λ.

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
