# Peer review of "All-Directional DOA Estimation for Ultra-Wideband Regular Tetrahedral Array Using Wrapped PDoA"

_sensors, 2022, doi:10.3390/s22041532_

Round 1
Reviewer 1 Report
The paper proposed an efficient method for joint TDoA and wrapped PDoA localization for UWB array systems. The method seems taking advantage of the tetrahedral structure of the array.
1) The authors need to evaluate the complexity of the proposed algorithm compared to the state-of-the-art, such as that in [1].
2) Is the accuracy approaching the CRLB? Please compared the performance with the bound. Together with the previous comment, the authors should compare the performance with others'.
3) The simulation accuracy seems optimistic, and the authors should use more realistic parameters. Otherwise, the angular error of 0.0942 degree is useless in terms of practical guidance.
Author Response
Dear reviewer,
Thank you for your comments on our article. We responded to your comments and modified the article correspondingly.
- The computational complexity of our proposed algorithm is , M is the antenna number in a subarray. The computational complexity of a subarray is , the worst case searching complexity is , and every time computing a tetrahedron need compute 4 subarrays. More specifically, when subarray is 3-antenna UCA, ,the computational complexity is . As we know , the computational complexity of a RTA are dependent on RTA aperture and wavelength . Within the application of IOT or robotic location sensors, the reasonable size is . On the other hand, the wavelength of UWB band(3-8GHz) is range from 0.1m to 0.0375m. In the worst case, the computational complexity is .The computational complexity of reference [1] algorithm 1 is , M=4. Their search area is a spherical surface of radius d. is the sphere triangle mesh points total number. Ns is the number of sampling points, is SPI total number. But, the article did not statement those parameters exactly. We do not know the minimum number of sampling points that can preserve the geometric features and local optimums of their cost function. We can only speculate on those parameters. is related to sampling accuracy of P. when P is accurate in centimeters , ; when P is accurate in millimeters , . Therefore, computational complexity of reference [1] algorithm 1 is relatively larger than ours.
In the terms of calculating time, a loose comparison is made: ours algorithm running on a Stm32F4 series 168MHz MCU in 20Hz, while reference [1] algorithm 1 costs 229 us on a 2.3GHz CPU, and PSO algorithm in reference[21] costs 3.9 s on a 2.3GHz CPU.
- We added CRLB curves on Figure 7. The accuracy of our proposed algorithm is approaching closely to CRLB with different And our results are consistent with those in reference [14][18]. When ambiguities estimated correctly, the results accuracy coincide with phased UCA.
- In Figure 7, the horizontal axis means both TDoA and PDoA SNRs. The convergence performance of the proposed method depends not only on PDoA SNR but also on TDoA SNR. A lower TDoA SNR means the initial values may be out of the Nmax range, and result in search failure. Therefore, different will have different effects on the convergence of the proposed algorithm. For example, when , SNR lower than 20dB, there is no guarantee that every search will be a success. In order to draw the different ratios together we chose 20dB-40dB range. If release TDoA SNR, a guaranteed search success will be made in a lower PDoA SNR condition, even approach 0dB. The search success boundary with TDoA SNR and PDoA SNR was discussed numerically in Figure 9.
In the application scenario of lower speed and short range localization, 0.1degree angular error is acceptable. Take UAV takeoff/ landing for example. The landing zone is 1m by 1m square, where deployed a UWB beacon. A RTA sensor is deployed on a drone. Horizontal positioning error . When the altitude ,;and when the altitude ,. Obviously, the horizontal error is decreased by descending, and , technically, this accuracy can meet the needs of UAV landing. Compared with the existing positioning method, GNSS provides 3-5M positioning accuracy, RTK GNSS provides 5-20cm positioning accuracy, our technology has obvious advantages.
Best Wishes!
Sincerely yours
Jinglin Luo
Reviewer 2 Report
I'm impressed by the development of the theory and the method, and comparison between the proposed method and others to show the strengths of the method they present.
One thing I'm concerned with is the use of 'Ultra-Wideband' from the title to the end.
This method seems general and 'Ultra-Wideband' may be omitted.
Author Response
Dear reviewer,
Thank you for your high comments on our article.
Our research work began with the development of robot positioning system based on UWB. Since UWB can measure both TDoA and PDoA at the same time, we develop the direction finding algorithm proposed in the paper. Considering that our existing technology is only limited in the field of UWB direction finding and location, so we titled the method proposed in this paper remain ‘Ultra-Windeband’. Whether the method proposed in this paper is applicable to more general fields still needs to be verified in the following work.
Thanks again for your affirmation of our work.
Best Wishes!
Sincerely yours
Jinglin Luo